# Vegan diet in young children remodels metabolism and challenges the statuses of essential nutrients

Topi Hovinen[1,†] [ID], Liisa Korkalo[2,†] [ID], Riitta Freese[2] [ID], Essi Skaffari[2] [ID], Pirjo Isohanni[1,3] [ID], Mikko Niemi[4,5] [ID], Jaakko Nevalainen[6] [ID], Helena Gylling[7] [ID], Nicola Zamboni[8] [ID], Maijaliisa Erkkola[2] [ID] & Anu Suomalainen[1,5,9,*] [ID]

## Abstract

**Vegan diets are gaining popularity, also in families with young children. However, the effects of strict plant-based diets on metabolism and micronutrient status of children are unknown. We recruited 40 Finnish children with a median age 3.5 years—vegans, vegetarians, or omnivores from same daycare centers—for a cross-sectional study. They enjoyed nutritionist-planned vegan or omnivore meals in daycare, and the full diets were analyzed with questionnaires and food records. Detailed analysis of serum metabolomics and biomarkers indicated vitamin A insufficiency and border-line sufficient vitamin D in all vegan participants. Their serum total, HDL and LDL cholesterol, essential amino acid, and docosahexaenoic n-3 fatty acid (DHA) levels were markedly low and primary bile acid biosynthesis, and phospholipid balance was distinct from omnivores. Possible combination of low vitamin A and DHA status raise concern for their visual health. Our evidence indicates that (i) vitamin A and D status of vegan children requires special attention; (ii) dietary recommendations for children cannot be extrapolated from adult vegan studies; and (iii) longitudinal studies on infant-onset vegan diets are warranted.**

**Keywords** development; metabolism; nutrition; vegan; vitamin

**Subject Category** Metabolism

See also: **AE Allen & JW Locasale** (February 2021)

## Introduction

A vegan diet is gaining popularity among Western societies, and a growing number of children are born from vegan mothers (Baldassarre *et al*, 2020). However, knowledge on the metabolic consequences of a strict vegan diet in infants and children is scarce which may currently lead to conflicting views between healthcare professionals and families in need of healthcare interventions (Farella *et al*, 2020).

Abstaining from any food of animal origin can be motivated by ecological, ethical, health-related, or religious reasons (Janssen *et al*, 2016). Positive health effects of vegan diets include lower body mass index (BMI), non-HDL cholesterol, and fasting blood glucose, based on studies on vegan adults, or conclusions extrapolated from studies on vegetarians (Dinu *et al*, 2017). Adults following a vegan diet have been reported to have a reduced risk for ischemic heart disease, type-2 diabetes, and all cancers combined, but an increased risk for bone fractures and brain hemorrhages (Appleby & Key, 2016; Tong *et al*, 2019). Furthermore, metabolic profiling in adults indicated that consuming a vegan diet results in a distinct metabolomics footprint (Schmidt *et al*, 2015; Schmidt *et al*, 2016). The nutrients considered as potentially critical include protein, vitamin B12, iodine, vitamin D, calcium, iron, zinc, long-chain n-3 fatty acids, riboflavin, and vitamin A (Kristensen *et al*, 2015; Schürmann *et al*, 2017; Baldassarre *et al*, 2020). Perhaps surprisingly, studies in infants and children are limited to anthropometric studies suggesting diminished average growth, yet still within normal range, and case reports of life-threatening micronutrient deficiencies from poorly planned vegan diets (Sanders, 1988; Pawlak, 2017).

Children require more energy and nutrients per body weight unit than adults to ensure normal growth and development of neural, endocrine, and immunological systems (Heird, 2012). Hence, dietary recommendations for children cannot be extrapolated of conclusions from adult vegan studies. Furthermore, studies on

1  Research Programs Unit, Stem Cells and Metabolism, University of Helsinki, Helsinki, Finland
2  Department of Food and Nutrition, University of Helsinki, Helsinki, Finland
3  Department of Pediatric Neurology, Children's Hospital, University of Helsinki and Helsinki University Hospital, Helsinki, Finland
4  Individualized Drug Therapy Research Program, University of Helsinki, Helsinki, Finland
5  HUSLAB, Helsinki University Hospital, Helsinki, Finland
6  Health Sciences, Faculty of Social Sciences, Tampere University, Tampere, Finland
7  Department of Medicine, Division of Internal Medicine, University of Helsinki, Helsinki, Finland
8  Institute of Molecular Systems Biology, ETH Zürich, Zürich, Switzerland
9  Neuroscience Center, HiLife, University of Helsinki, Helsinki, Finland
   *Corresponding author. Tel: +358 9 4717 1965; E-mail: anu.wartiovaara@helsinki.fi
   †These authors contributed equally to this work as first authors

vegetarian children provide limited information on vegans, as including animal-based foods in the diet shifts the intake of several nutrients and the plasma metabolic profile significantly toward those of omnivores (Schmidt et al, 2015).

Finland is a high-income country with public daycares of excellent quality, available for the families for the six first years of life of children. Up to 75% of Finnish children attend public daycares (Haapamäki & Ranto, 2015), which offer standardized nutritionist-planned daily meals free of charge (Korkalo et al, 2019). Such a uniform system offers an excellent opportunity to study effects of vegan diet in young children. Here, we use targeted and untargeted metabolomics approach to comprehensively compare the nutritional and metabolic statuses of Finnish daycare children following vegan and omnivore diet.

# Results

## Participants

All vegan participants had followed a vegan diet since birth and were breastfed for 13–50 months by vegan mothers. All vegan participants had been weaned from breastfeeding more than a year before the study, and none of the participants were breastfed at the time of the study. All participants were of Finnish origin and were apparently healthy with no reported systemic medication according to the questionnaires. Table 1 presents the characteristics of the participants.

## Anthropometrics and diet

Figure 1A illustrates the heights and BMIs of the participants compared to the current Finnish growth references. No differences were indicated between the diet groups in the z-scores of height, BMI, or mid-upper arm circumference (Table 1). The children on the

vegan diet had lower intake (percentage of energy, E%) of protein and saturated fatty acids, and higher intake of mono- and polyunsaturated fatty acids than the omnivorous children (Fig 1B, Table EV1). The calculated intakes of linoleic acid (LA) and alpha-linoleic acid (ALA) were higher in vegans than in omnivores. The vegan diets included only trace amounts of cholesterol, and no eicosapentaenoic acid (EPA) or docosahexaenoic acid (DHA). Fiber and folate intakes were higher in vegans than in omnivores. Vitamin B12 intake was similar in the diet groups, with the main food sources for vegans being fortified drinks and brewer's yeast flakes. We found no difference in the total intake of vitamin A in retinol activity equivalents (RAE) between the groups. The only foods containing retinoids in the reported vegan diets were margarines fortified with vitamin A, which contributed 33% of the total RAE in the diet of this group. The corresponding proportions in vegetarians and omnivores were 41 and 60%, respectively. Most participants, including all vegans, took vitamin D supplements, and all but one vegan child took vitamin B12 supplements (Table EV1). Main food sources of energy and macronutrients are presented in Tables EV2–EV5.

## Cholesterol metabolism

The plasma total cholesterol, LDL cholesterol (LDL-C), and HDL cholesterol (HDL-C) concentrations were significantly lower in vegans than in omnivores. The cholesterol absorption biomarkers showed higher values in vegans than in omnivores, while there was no difference in cholesterol biosynthesis biomarkers (Fig 2B, Table EV6).

## Transthyretin and micronutrient biomarkers

Serum concentrations (Fig 2A, Table EV6) of transthyretin, RBP, 25-hydroxyvitamin $D_3$ (25(OH)$D_3$), and total-25(OH)D were lower in vegans than in omnivores. The RBP of all children in the vegan diet group fell below the insufficiency cut-off level, and the levels of

**Table 1. Characteristics of the participants.**

| | Omnivore (OMN) | Vegetarian (VGTR) | Vegan (VGN) | P-value[†] | | | Total |
|---|---|---|---|---|---|---|---|
| | | | | VGN vs OMN | VGN vs VGTR | VGTR vs OMN | |
| n | 24 | 10 | 6 | NA | NA | NA | 40 |
| Age – years[a] | 3.89 [1.42 to 7.07] | 3.37 [1.73 to 5.67] | 3.32 [1.75 to 6.34] | NA | NA | NA | 3.96 [1.42 to 7.07] |
| Sex – F:M | 12:12 | 4:6 | 3:3 | NA | NA | NA | 19:21 |
| Number of sibling pairs | 0 | 1 | 1 | NA | NA | NA | 2 |
| Height – z-score[b] | −0.44 [−3.52 to 1.09] | −0.88 [−2.44 to 0.41] | 0.04 [−1.03 to 0.69] | 0.48 | 1.00 | 0.13 | −0.46 [−3.52 to 1.10] |
| BMI – standard deviation score[b] | 0.27 [−2.59 to 3.06] | 0.54 [−1.18 to 1.83] | −0.05 [−1.57 to 2.15] | 0.88 | 0.11 | 0.37 | 0.40 [−2.59 to 3.06] |
| MUAC – z-score[c] | 0.24 [−0.96 to 1.17] | 0.24 [−0.87 to 1.26] | 0.17 [−0.84 to 2.07] | 0.85 | 0.50 | 0.91 | 0.22 [−0.96 to 2.07] |

Statistical distributions are described as medians [minimum – maximum]. BMI = body mass index, MUAC = Mid-Upper Arm Circumference.
[a]Age is presented as that at the time of anthropometric measurements.
[b]Z-score for height and SDS-score for BMI was calculated from Finnish population growth data (Saari et al, 2011).
[c]Z-score for mid-upper arm circumference (MUAC) was calculated from the WHO arm circumference-for-age growth standards for 1- to 5-year-olds and from extended growth standard tables by Mramba et al (2017) for children older than 5 years old (World Health Organization, 2007).
[†]P-values for pairwise comparisons between vegans and omnivores (primary analysis) and between vegetarians and other dietary groups (secondary analysis) were calculated using age- and sex-adjusted exact permutation tests with n = 47,500 permutations. NA = not applicable.

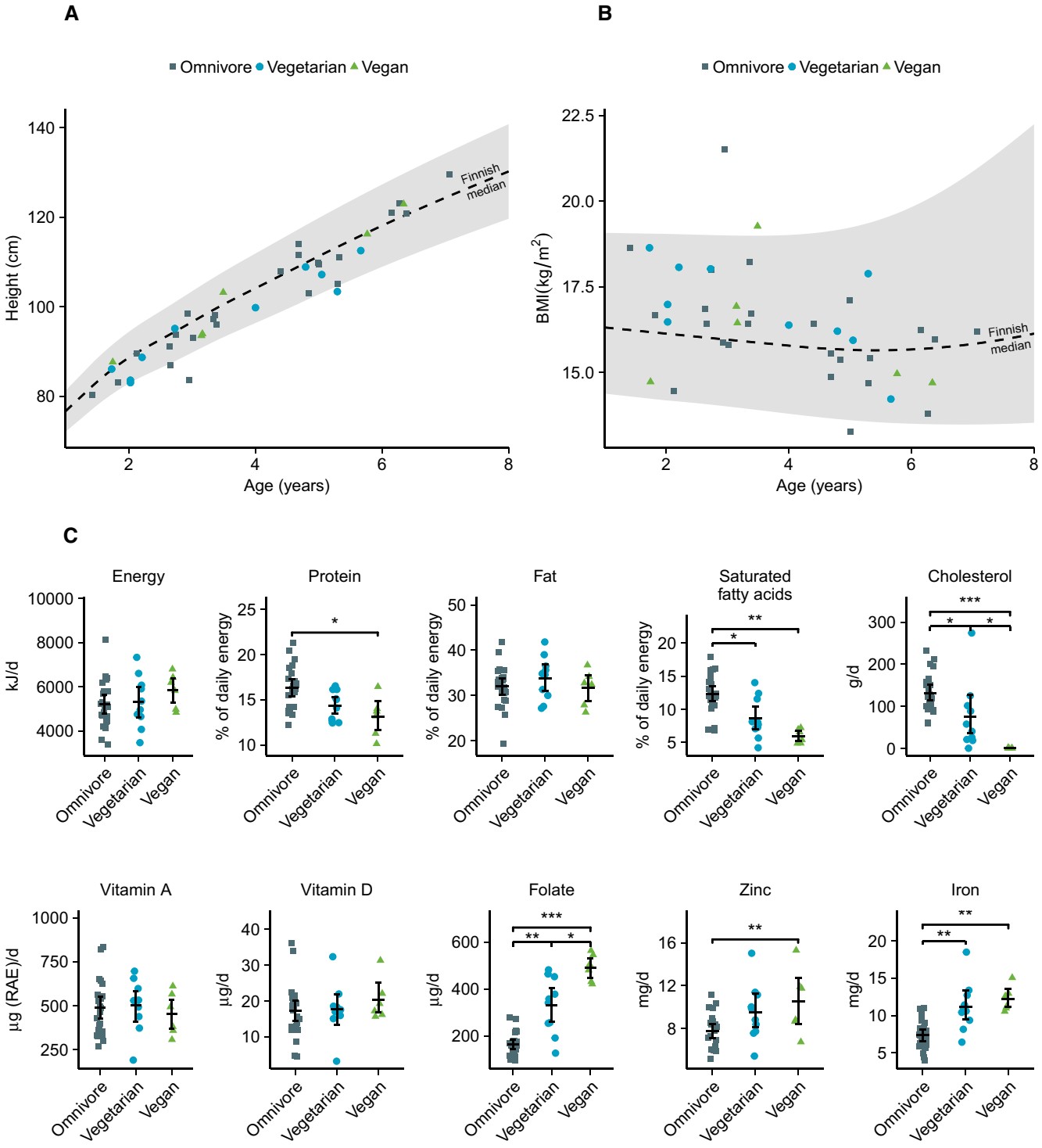

**Figure 1. Anthropometric measurements and dietary intake of omnivores, vegetarians, and vegans.**

A, B  Height (A) and body mass index (B) of the participants by age in the diet groups compared to 2 SD intervals (shaded area) and median in the Finnish reference population.

C  Dietary intake of selected macro- and micronutrients of participants.

Data information: The summary bar is presented as mean ± SEM. Differences between diet groups were evaluated with age- and sex-adjusted permutation tests with Benjamini–Hochberg correction for multiple testing. Only significances $P < 0.05$ are displayed. $n = 24$ omnivores, 10 vegetarians, six vegans. $*P < 0.05$, $**P < 0.01$, $***P < 0.001$. Exact $P$-values are provided in Table EV1.

Source data are available online for this figure.

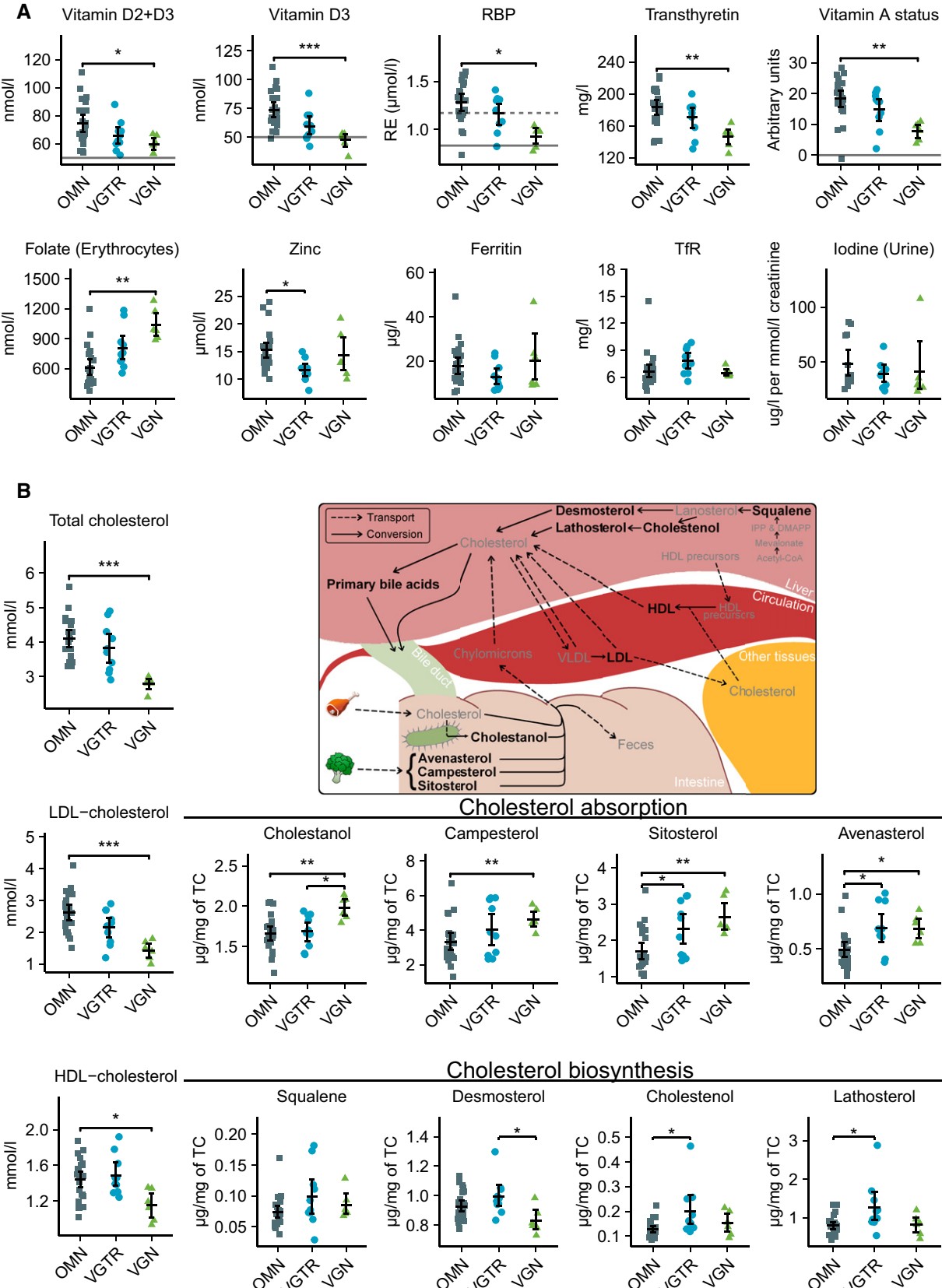

**Figure 2.**

**Figure 2.   Biomarkers of micronutrient statuses and cholesterol metabolism in omnivores, vegetarians, and vegans.**

A   Serum, plasma, or urine levels of vitamin D, vitamin A, folate, zinc, iron, and iodine status biomarkers in diet groups. Solid lines indicate cut-offs for deficiency, and dashed lines indicate cut-offs for insufficiency. The RBP limits for vitamin A insufficiency and deficiency are validated for the method at 1.17 and 0.83 μmol/l, respectively. Vitamin A status was calculated based on RBP, transthyretin, and CRP (Talsma *et al*, 2015).
B   Cholesterol metabolism biomarkers in the diet groups. The subfigure summarizes cholesterol metabolism in humans and highlights cholesterol-related metabolites measured in this study with black font (Risley, 2002; Nelson & Cox, 2012).

Data information: The summary bar is presented as mean ± SEM. Differences between diet groups were evaluated with age- and sex-adjusted permutation tests with Benjamini–Hochberg correction for multiple testing. Only significances $P < 0.05$ are displayed. $n = 24$ omnivores, 10 vegetarians, six vegans except for panel (A) iodine, where $n = 13$ omnivores, nine vegetarians and six vegans. *$P < 0.05$, **$P < 0.01$, ***$P < 0.001$. Exact *P*-values are provided in Table EV6. DMAPP, dimethylallyl pyrophosphate; HDL, high-density lipoprotein; IPP, isopentenyl pyrophosphate; LDL, low-density lipoprotein; OMN, omnivore; RBP, retinol-binding protein; VGN, vegan; VGTR, vegetarian; VLDL, very low-density lipoprotein.
Source data are available online for this figure.

two of the children fell below the deficiency cut-off level, while the regression model of vitamin A status classified only one omnivore as vitamin A deficient. Vegans had more erythrocyte folate than omnivores. We found no differences between vegans and omnivores in serum ferritin, transferrin receptor, zinc, or spot urine sample iodine concentrations. Secondary statistical analysis between all groups suggests a lower zinc concentration in the vegetarian group than in omnivores. Serum concentrations of transcobalamin-bound vitamin B12 were adequate in all groups. Four (67%) vegans, nine (90%) vegetarians, and 22 (92%) omnivores had transcobalamin-bound vitamin B12 above the linear detection limit of 128 pmol/l. The smallest measured concentration was 77 pmol/l, and the cut-off for further clinical examination is 70 pmol/l.

### Bile acid metabolism

Pathway analysis from untargeted metabolomics (Fig 3A, Appendix Table S3) highlighted bile acid synthesis as highly different in vegans compared to omnivores. Direct investigation of serum bile acids revealed higher steady-state levels of unconjugated primary bile acids and a lower taurine to glycine conjugation ratio of bile acids in vegans than in omnivores (Fig 3B, Table EV6). Serum levels of bile acids in total (Appendix Table S2) and bile acid synthesis biomarker 7-alpha-hydroxy-4-cholesten-3-one (Table EV6) did not differ among diet groups.

### Amino acids and fatty acids

Untargeted metabolomics indicated a pattern of overall lower concentrations of circulating essential amino acids (Fig 3C) in vegans. Only a few individual amino acids showed significant differences. The largest differences in essential amino acids were seen in branched-chain amino acid levels. Analysis of fatty acid compartments (Appendix Table S4) in serum indicated lower levels of DHA, higher levels of ALA (Fig 4A) and long-chain fatty acid carnitines (Fig 4B), higher lysophosphatidylcholine (lysoPC) per lysophosphatidylethanolamine (lysoPE) ratio (Fig 4C), and higher level of triglycerides with total carbon atom number correlating to medium-chain fatty acid tails (Fig 4D) in vegans than in omnivores.

### Hierarchical clustering

Hierarchical clustering (Fig EV1) of participants based on the untargeted metabolomics data resulted in four major clusters. 80% of vegan participants (5/6) clustered together, forming 56% (5/9) of

cluster B. Vegetarian participants were most dispersed among clusters with 4/10 vegetarians forming 44% of cluster B and 6/10 vegetarians clustering together with most of the omnivores to clusters A and D.

## Discussion

Here, we report that diet markedly modifies the metabolism of young children. The sample was homogenous and unique: The children were of Finnish origin, had a median age of less than four years, and consumed meals that were centrally planned to fulfill dietary recommendations. The children who followed the vegan diet from birth showed a metabolic profile and nutrient status distinct from those of lacto-ovo-vegetarians and omnivores, indicating that only relatively little animal source foods are enough to shift the metabolism of children. The main findings in vegan children included very low cholesterol concentrations and modified bile acid metabolism, as well as their markedly low fat-soluble vitamin status despite their nutrient intakes matching current national recommendations fairly well. Despite of the adequate estimated vitamin A intake, the RBP results of vegan children in our sample indicated insufficient vitamin A status. Their vitamin D levels were low although the samples were taken during and after summer with expectedly high sunlight exposure and vitamin D storage. Our evidence indicates that special attention is needed to ensure adequate status of these important micronutrients for children on a vegan diet.

Children on a vegan diet showed strikingly low plasma HDL-C and LDL-C as well as total cholesterol levels, with a median total cholesterol level of 2.85 mmol/l. The value was markedly lower than the median total cholesterol level of 3.7 mmol/l in Finnish adults following a vegan diet (Elorinne *et al*, 2016). Low non-HDL cholesterol in vegans has been reported in different studies (Elorinne *et al*, 2016; Benatar & Stewart, 2018). This may reflect the cholesterol-lowering elements (Mach *et al*, 2019) in well-planned vegan diets such as the negligible amount of dietary cholesterol, the dietary fatty acid profile that is low in saturated fatty acids and high in unsaturated fatty acids, and a high fiber intake. The few children in our sample with high LDL-C and total cholesterol belonged to the omnivore group. The endogenous hepatic cholesterol biosynthesis markers were similar between the vegan and omnivore children. These data suggest that endogenous cholesterol biosynthesis does not show a compensatory response to lack of dietary cholesterol.

The low cholesterol levels resulting from adult vegan diet have mostly been linked to positive cardiovascular health effects

(Appleby & Key, 2016; Elorinne *et al*, 2016), although a recent study also suggested an increased risk for stroke (Tong *et al*, 2019). The markedly low cholesterol in vegan infants and children in our study raises the question of whether such levels are healthy, as cholesterol is essential for cellular growth, division, and development of physiological systems due to its major role in the synthesis of cell membranes, steroid hormones, bile acids,

and brain myelin. Early studies on LDL receptors suggested that the physiological concentration of blood LDL-C may be as low as 0.65–1.6 mmol/l (vegan children in our study ranged from 1.0 to 1.8 mmol/l) (Brown & Goldstein, 1986; O'Keefe *et al*, 2004). However, longitudinal studies on the health effects of consuming a strict vegan diet since birth have not been conducted.

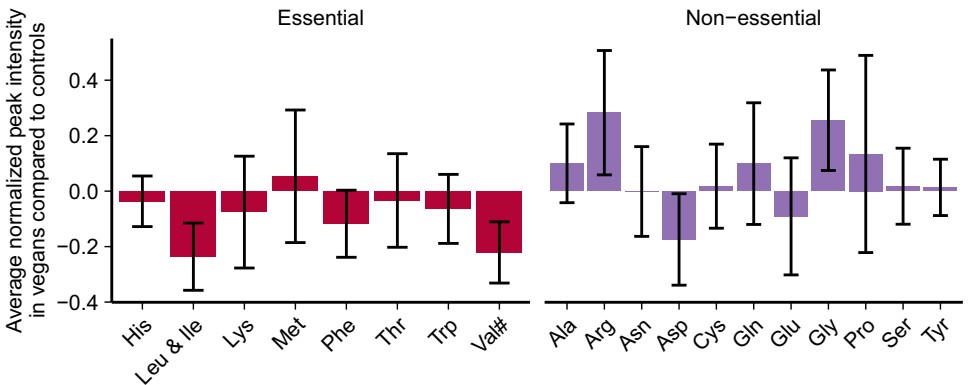

**Figure 3.**

**Figure 3.  Analysis of untargeted flow injection TOF-MS metabolomics data and further targeted analysis of serum bile acid concentrations.**

A   Pathway analysis from 872 detected untargeted metabolites between omnivores and vegans.
B   Targeted analysis of bile acid concentrations in serum of omnivores, vegetarians and vegans.
C   Amino acid levels in untargeted metabolomics. Numeric means and standard deviations in this figure can be found in Appendix Table S4. [#]Valine and betaine are two major components under the same *m/z* peak, causing uncertainty in the interpretation of valine levels.

Data information: Pathway analysis in subfigure (A) was performed with a gene-set enrichment analysis (GSEA)–based method. The summary bar of subfigures (B, C) is presented as mean ± SEM. Differences between diet groups in subfigure (B) were evaluated with age- and sex-adjusted permutation tests with Benjamini–Hochberg correction for multiple testing. Only significances $P < 0.05$ are displayed. $n = 24$ omnivores, 10 vegetarians, six vegans in panel (B). *$P < 0.05$. Exact *P*-values for panel (A) are provided in Appendix Table S2 and for panel (B) in Table EV6. ALA, alpha-linolenic acid; BCAA, branched-chain amino acid; BA, bile acid; LA, linoleic acid; OMN, omnivore; VGTR, vegetarian; VGN, vegan, VLCFA, very long-chain fatty acid.

Source data are available online for this figure.

The main route of cholesterol excretion from the body is through bile acids, the biosynthesis of which occurs in the liver. Our metabolomics analysis indicated that bile acid biosynthesis was the pathway that differed most significantly between the diet groups. In vegans, direct measurement revealed higher primary bile acids, cholic acid, and chenodeoxycholic acid, which were previously reported to increase upon fasting in children (Barbara *et al*, 1980), and a lower taurine to glycine ratio in bile salt conjugation than omnivores. Vegan diets contain only little taurine, and the relatively low taurine-conjugation compared to glycine conjugation of bile salts in vegan children is in accordance with previous adult studies (Ridlon *et al*, 2016). In addition to the role of bile acids in digestion and absorption of fat-soluble components from the diet, recent studies have elucidated their diverse roles in endocrine and metabolic signaling and gut–microbiome–brain interactions (De Aguiar Vallim, 2013; Ridlon *et al*, 2016; Kiriyama & Nochi, 2019). What physiological consequences such findings indicate in children following a strict vegan diet remains to be studied. Our evidence indicates that vegan diet remarkably modifies bile acid homeostasis in young children.

The biomarkers for fat-soluble vitamins A and D showed markedly low levels in the Finnish children following a vegan diet, although there were no indications of compromised absorption of fat-soluble dietary compounds. The total fat intake in vegan group was similar, and cholesterol absorption biomarkers showed higher levels than those of omnivores. Vitamin D insufficiency is a well-established concern in Northern countries with restricted exposure to sunlight (Itkonen *et al*, 2020). The seasonal variation was observed in vitamin D status in Danish children from 2 to 14 years of age. The high peak levels in autumn were between 11 and 19 nmol/l higher than during the lowest season in spring for supplement users and slightly greater for non-supplemented individuals (Hansen *et al*, 2018). Vegan children in our sample had lower status of vitamin D than omnivores despite all vegan families reporting daily use of supplements that reached the daily vitamin D intake recommendations (THL, 2019), and the blood samples having been collected during the high peak of seasonal variation in vitamin D status. Different forms of vitamin D fortification may play a role in low status of vitamin D in vegan children. Vegan supplements contain "vegan-friendly" vitamin $D_3$, whereas vegan food products, such as soymilk, are often fortified with vitamin $D_2$. Vitamin $D_3$ has been suggested to be more effective than $D_2$ at raising total 25(OH)D concentrations, especially in the wintertime (Tripkovic *et al*, 2017). The vegan children in our study had levels of the endogenous and animal-based form D3 between 33 and 53 nmol/l, and total vitamin D between 53 and 67 nmol/l, when the clinical cut-off of insufficiency of total vitamin D level 50 nmol/l. Additionally, lower

vitamin A intake in vegan adults has been suggested previously (Kristensen *et al*, 2015). The calculated intake of vitamin A in the different diet groups of our sample was similar. Despite this similarity, based on the RBP levels reflecting the actively available vitamin A, the vitamin A status of all vegans was insufficient and in two vegan children RBP concentrations were below the deficiency cut-off. Notably, RBP is considered reliable in group level analysis of vitamin A status and the assessment on individual level has some pitfalls (Tanumihardjo *et al*, 2016). The linear model for vitamin A status considering inflammatory status did not classify any vegans as vitamin A deficient, but showed significantly lower status for vegans than omnivores, in agreement with RBP alone. RBP synthesis shows complex regulation together with hepatic vitamin A, zinc and iron levels, and overall protein and energy intake (Tanumihardjo *et al*, 2016). According to our data, the energy intake and zinc and iron status did not differ between vegans and omnivores. Lower protein intake, transthyretin levels, and essential amino acid levels in vegans compared to omnivores may affect the protein status in vegans and therefore the interpretation of RBP levels as vitamin A biomarker. Our results indicate, however, that the vitamin D and A statuses of children following a vegan diet require special attention. Direct measurements of serum retinol, clinical measurement of vitamin A status such as dark adaptation tests and comparison of vegan vitamin D status at winter season are required for further evaluation of vitamin A and D statuses in vegan children.

The vitamin B12, zinc, iron, and iodine statuses, previously found to be challenged in adult vegans (Craig, 2009; Elorinne *et al*, 2016), did not differ between the diet groups. Intakes of zinc and iron were in fact significantly higher in vegans than in omnivores. Vegans had higher folate intake and concentration than omnivores, and four out of six vegans had levels above the reference range 208–972 nmol/l. Although high folate status is traditionally considered to have positive health effects, recent studies have raised concerns on possible adverse effects of high folate status combined to low vitamin B12 status on neurocognitive health and birth outcomes (Maruvada *et al*, 2020).

The dietary data of vegan children in our sample indicated protein intake of 10–16 E%, which is in line with recommendations (THL, 2019). However, the untargeted metabolomics suggested that their overall circulating essential amino acid pools were systematically lower than those of omnivores, specifically those of branched-chain amino acids. Similar findings have been reported in adult vegans (Schmidt *et al*, 2016; Lindqvist *et al*, 2019). Serum transthyretin has a short half-life and is sensitive to the availability of essential amino acids and vitamin A in the liver (Dellière & Cynober, 2017). The transthyretin concentration was also lower in vegans

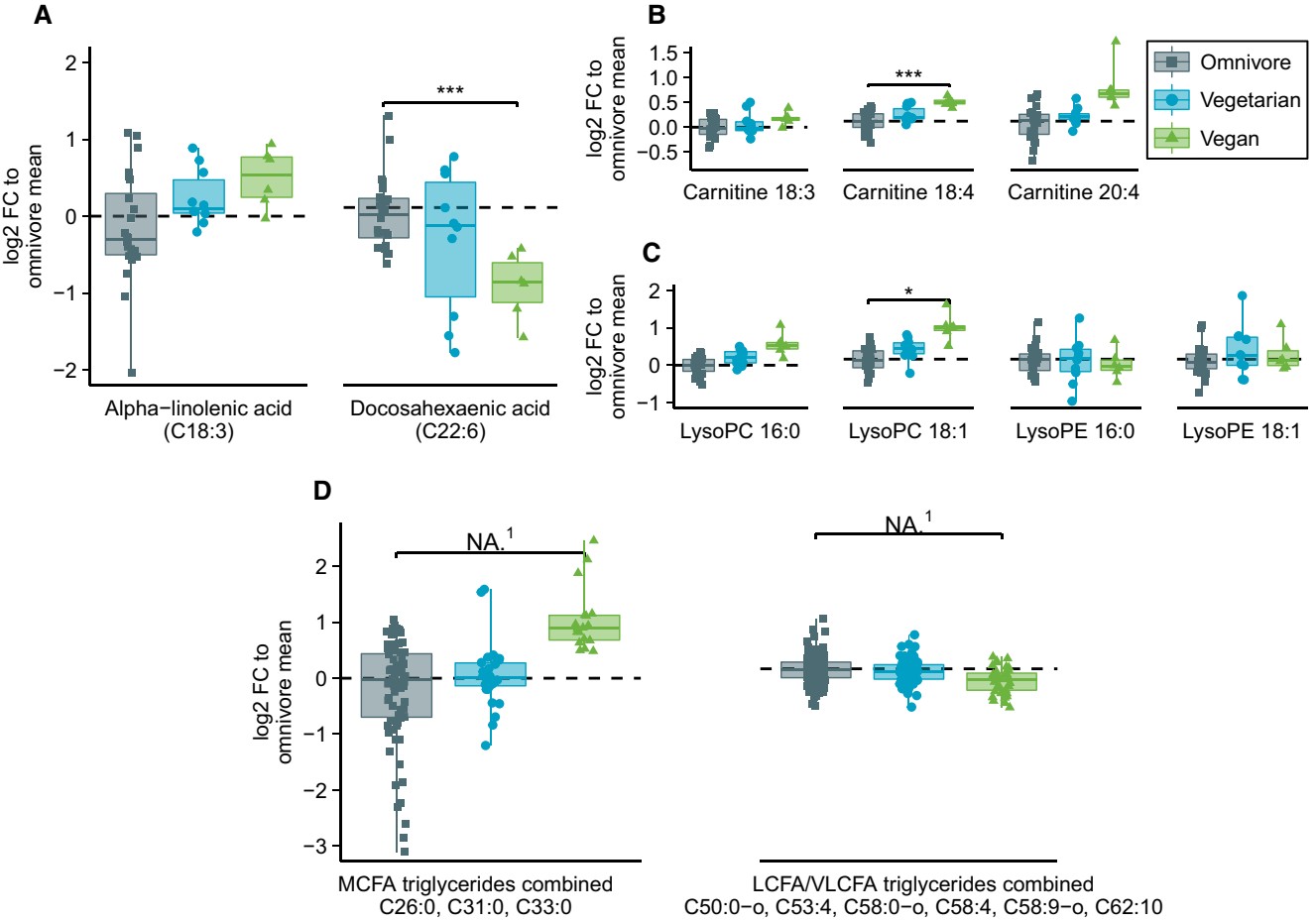

**Figure 4. Fatty acid analysis from untargeted MS metabolomics in omnivores, vegetarians, and vegans.**

A   Fatty acid analysis from untargeted MS metabolomics including free alpha-linolenic acid (ALA) and docosahexaenoic acid (DHA).
B   Carnitine-bound fatty acids commonly found in serum that were detected in mass spectrometry.
C   Lysophosphatidylcholine and lysophosphatidylethanolamine with C16:0 (palmitic acid) and C18:1 (oleic acid).
D   Combined analysis of all found triglycerides corresponding to medium-chain fatty acid lengths and long or very long-chain fatty acid lengths.

Data information: Box plot center represents group median, hinge covers 25th to 75th percentile and whiskers cover interval from minimum to maximum. Zero-level represents omnivore mean to which each individual is compared in $\log_2$ scale. Differences between diet groups in subfigures (A–C) were evaluated with Student's *t*-test (unequal variance, two-sided) with Benjamini–Hochberg correction for multiple testing. Only significances $P < 0.05$ are displayed. Exact *P*-values are provided in Appendix Table S8. In subfigures (A–C), $n = 24, 10, 6$ for omnivores, vegetarians, and vegans, respectively. In subfigure (D), multiple-related metabolites (three MCFAs and six LCFAs/VLCFAs) are combined to same box plot yielding $n = 72, 30$, and 18 in MCFA subfigure and $n = 144, 60$, and 36 in (V)LCFA subfigure for omnivores, vegetarians, and vegans, respectively. *$P < 0.05$, ***$P < 0.001$, NA, not applicable. Cxx:y, fatty acid of xx carbon atoms and y double bonds; FC, fold change; LysoPC, lysophosphatidylcholine; LysoPE, lysophosphatidylethanolamine; MCFA, medium-chain fatty acid; LCFA, long-chain fatty acid; VLCFA, very long-chain fatty acid. (1) Student's *t*-test is not suitable for testing of multiple dependent metabolites in a single test. Of single metabolites included in subfigure (D), Benjamini–Hochberg adjusted *t*-test was significant ($\alpha = 0.05$) for C50:0-o, C53:4, and C58:0-o.

Source data are available online for this figure.

than in omnivores, albeit still in the reference range. Further correlation analysis (Appendix Table S5) showed that branched-chain amino acids correlated positively to serum transthyretin levels, and lysine negatively with standardized MUAC. The source of different patterns of circulating amino acids in children is not well known. Increased circulating branched-chain amino acid concentrations are associated with obesity and the risk of insulin resistance in both adults and children (Zhao *et al*, 2016), whereas undernourished children show chronically low circulating essential amino acid concentrations (Semba *et al*, 2016). Our evidence of low transthyretin and essential amino acid levels invites attention to dietary

protein quality, not only proportional intake measured as E%, in growing children following a vegan diet. Follow-up studies, specifically focusing on amino acid quantities, will enlighten the aspect further.

Vegan diets are rich in the essential fatty acids ALA and LA, but practically devoid of the ALA derivatives DHA and EPA, long-chain n-3 fatty acids of which DHA is needed for visual process and synaptic functioning (Sanders, 2009). Accordingly, we found high intake of ALA and low intake of EPA and DHA in the diet vegan children. Untargeted metabolomics suggested consistent findings, high ALA and low DHA, in serum levels. This correlates well to findings in

vegan adults (Sanders, 2009). Vegan children have not been found to have compromised declined visual function linked to primary DHA deficiency (Sanders, 2009). However, DHA and active vitamin A are both important for eyesight (Lien & Hammond, 2011), and the low statuses of both in children may raise a concern for the visual health.

Vegan children show widespread differences to omnivores and vegetarians also in other serum fatty acid compartments. In accordance with our results, higher circulating long-chain fatty acid carnitine levels, and higher lysoPC/lysoPE ratio have earlier been associated with diets with lower dairy intake and higher unsaturated/saturated fat ratio (Playdon *et al*, 2017). Recent studies have increased our knowledge on the signaling potential of circulating lysophospholipids (Makide *et al*, 2014). The intracellular role of carnitines and medium-chain fatty acids are well known for mitochondrial energy production (Schönfeld & Wojtczak, 2016). However, the current understanding on the roles and significance of extracellular circulating different fatty acid carriers for health is scarce and particularly insufficient in children.

The unsupervised hierarchical clustering of untargeted metabolomics data indicated the clustering of vegans separate from omnivores, indicating the major effect of diet to metabolism of healthy children. However, the vegetarians showed heterogeneous clustering, 60% clustering with omnivores, and the rest with the vegans. Most of the measured biomarkers demonstrated a similar but more subtle trend in the vegetarian group than in vegans compared to omnivores. Our vegetarian group consisted of children who consumed fully vegan meals in daycare and pesco-/lacto-ovo-vegetarian diet at home. In full-time care, the daycare meals of Finnish daycare children account for approximately 50–60% of the daily intake of energy and most of the macro- and micronutrients during weekdays (Korkalo *et al*, 2019). The evidence indicates that even part-time consumption of lacto-ovo-vegetarian products in an otherwise strict vegan diet may substantially alleviate the risk to nutrient deficiencies in children. Our data indicate the importance of studying vegan children to enable evidence-based nutritional recommendations.

To conclude, our study demonstrates exceptional clustering of metabolic readouts in different diet groups of young Finnish children, enjoying centrally planned daycare diets designed to meet dietary requirements. Our data of lower status of several biomarkers in vegan children compared to omnivores, in the relatively low number of study subjects, calls for larger studies before early-life vegan diet can be recommended as a healthy and fully nourishing diet for young children, despite its many health-promoting effects in adults. We suggest that the metabolic effects of vegan diet in adults cannot be generally extrapolated to children. Long-term follow-up studies are needed to clarify the causes and consequences of lower levels of vitamin D, RBP, transthyretin, essential amino acids, total cholesterol, and DHA in vegan children.

# Materials and Methods

### Study design and participants

This study was approved by the coordinating ethics committee of Helsinki University Hospital and followed the guidelines of the Declaration of Helsinki and the U.S. Department of Health and Human Services Belmont Report. Parents of each child provided written informed consent.

This cross-sectional study was conducted while the City of Helsinki initiated a trial to provide a vegan diet option in 20 of its municipal daycare centers. Daycare centers offered an omnivore/vegan breakfast, lunch, and afternoon snack 5 days a week using standardized recipes. All menus were planned by a nutritionist to ensure compliance with dietary recommendations (THL, 2019).

During the recruitment period (flow chart in Appendix Fig S1) between May and October 2017, out of the 20 daycares with the vegan option, the parents of 31 children at 9 daycare centers had chosen a vegan diet for their child. Invitation letters were given to parents of all the children on vegan diet at daycare. We asked the daycare personnel of the same nine daycares to provide an invitation letter to parents of 161 children, who were selected based on the criteria that they were not receiving any special diet at daycare and they belonged to the same age group as the recruited vegan children. Parents of 25 children (81% of invited) receiving a vegan diet at daycare and 34 omnivore children (21% of invited) agreed to participate.

Our original plan was to study vegans and omnivores only. However, questionnaire data revealed that most participants consuming vegan food at daycare consumed either a vegetarian or an omnivorous diet at home. Therefore, we formed three diet groups: (i) vegans, (ii) "vegetarians" including lactovegetarians, lacto-ovo-vegetarians, and pescatarians who consumed vegan food at daycare, and (iii) omnivores. We excluded children who followed an omnivorous diet at home but a vegan diet at daycare because these participants ($n = 2$) did not fit into any of the diet groups. A further criterion for exclusion from the analysis was the lack of a blood sample. The sample for this article included 40 children (hereinafter participants): six strict vegans, 10 vegetarians, and 24 omnivores.

### Diet and anthropometry

The parents filled in questionnaires on dietary habits and basic health information and the dietary intake was calculated from an estimated 4-day food record filled in by the parents and daycare personnel between May and October 2017 (details in Appendix Supplementary Methods). The details of nutrient intake calculations are described in the Appendix Supplementary Methods. Anthropometric measurements were conducted according to the World Health Organization (WHO) procedures with two exceptions: Height was measured in the standing position for children under the age of two, and the same scale was used for children of all ages (WHO Expert committee, 1995). For height, we used Seca 213 portable stadiometer assembled and propped according to the manufacturer's instructions. No calibration rod was used. Weight was measured with Seca 878 Mobile flat scales and MUAC with Seca 201 circumference measuring tape.

### Blood and urine samples

Blood and urine samples were collected between June and October 2017. Venous blood samples of 15 ml were collected after overnight fasting and analyzed immediately in the clinical laboratory, the

remaining sample material was pseudonymized and cooled to 4°C, and the serum was separated and stored at −80°C.

Single random midstream urine samples were obtained from all participants by the families at home according to written instructions. The samples were pseudonymized and aliquoted to a 10 ml sterile trace element tube (Sarstedt item no. 62.554.502) and stored at −80°C.

### Blood and urine examinations

Standard laboratory tests and targeted analysis of micronutrients, inflammation biomarkers, blood lipids, glucose, and transthyretin were performed by accredited clinical laboratories. Serum cholestanol, plant sterols, and cholesterol biosynthesis intermediates were used as biomarkers of cholesterol absorption and biosynthesis and they were analyzed by gas–liquid chromatography (GLC) (Miettinen *et al*, 1990). Serum bile acids were analyzed by high-performance liquid chromatography–mass spectrometry (HPLC-MS/MS) (Xiang *et al*, 2010). Iodine and creatinine were measured in the urine samples of six vegans, nine vegetarians, and 13 omnivores. The samples for this analysis were chosen to cover whole age group in the study and equally include both sexes. We present two different cut-offs for non-sufficient RBP level, and we use the term insufficiency for the higher (1.17 μmol/l) and deficiency for the lower (0.83 μmol/l) limit (Tanumihardjo *et al*, 2016). Concentrations of transcobalamin II-bound vitamin B12 were measured up to 128 pmol/l. Appendix Table S1 presents a comprehensive list of methods with references, measurements, and clinical cut-offs.

### Untargeted metabolomics and lipidomics

Untargeted metabolomics analysis of serum samples was performed by flow injection-time-of-flight (TOF)-MS on an Agilent 6550 QTOF instrument in negative mode exactly as described by Fuhrer et al (Fuhrer *et al*, 2011). Putative annotations were created based on Human Metabolome Database v.3.6 using both accurate mass per charge (tolerance 0.001 *m/z*) and isotopic correlation patterns, allowing the detection of 872 metabolites. The approach is sufficient to assign molecular formulas in most cases, but does not distinguish between isomers. Full list of possible annotations of each detected metabolite is provided in Appendix Table S6. Untargeted metabolomics data were used in metabolic pathway analysis and analyses of amino acids and fatty acids that are considered abundant compared to alternative molecules matching the same *m/z* peak and that showed biologically relevant patterns in multiple metabolites in our data. The Appendix Supplementary Methods present the extraction protocols and comprehensive lists of the measured metabolites.

### Statistical analysis and figures

All statistical analyses and images were produced using R version 3.6.0 (code in Appendix Supplementary Methods). Anthropometric data were transformed to *z*-scores based on the Finnish growth standards (Saari *et al*, 2011), with the exception of mid-upper arm circumference, for which the WHO growth standards extended by Mramba *et al* (2017) were used (World Health Organization, 2007). Vitamin A status was assessed by retinol-binding protein

### The paper explained

#### Problem

The metabolic and nutritional consequences of a vegan diet in children are insufficiently known. We studied the metabolic and nutritional status and diet of Finnish daycare children between 1 and 7 years of age with vegan, vegetarian, and omnivore diets. The special value of the study is in the setting: The children enjoyed nutritionist-planned diets in daycares, designed to meet nutritional recommendations.

#### Results

Dietary assessment showed that the vegans had higher intake of folate and received smaller proportions of energy from protein and from saturated fatty acids than omnivores. Metabolomics and nutritional status biomarker analyses indicated that the statuses of transthyretin, essential amino acids, retinol-binding protein, vitamin D, docosahexaenoic acid, and cholesterol (including total, LDL and HDL) of vegan children were lower than those of omnivores. Their lipidomic and bile acid patterns were also distinct.

#### Impact

Our evidence indicates that vegan children show remarkable metabolic differences compared to omnivores. The data indicate that strict vegan diet affects metabolism of healthy children. Our study indicates the importance of detailed longitudinal and cross-sectional studies on the nutritional and health effects of a vegan diet before it can be recommended for infants or children.

(RBP) and a validated regression model of RBP, transthyretin, and C-reactive protein (CRP) (Talsma *et al*, 2015). Statistical analyses of targeted metabolites, intake, and anthropometrics between diet groups were conducted using age- and sex-adjusted permutation tests with probability index as test statistic. Statistical analyses of untargeted metabolomics were conducted using two-tailed Student's *t*-test assuming unequal variance. Benjamini–Hochberg correction was applied to take multiple comparisons in account. Comparisons between vegans and omnivores were considered as primary analyses, and comparisons of vegetarian group to omnivores and vegans were conducted as secondary analyses. To reveal the relation of vegetarian group to other two diet groups, hierarchical clustering of participants based on untargeted metabolomics data using agglomerative Ward's method with squared Euclidean distance as dissimilarity measure were conducted. Pathway analysis between omnivores and vegans was based on gene-set enrichment analysis (GSEA). Because untargeted metabolomics provides semi-quantitative results instead of concentrations, differences in individual amino acids and fatty acids were analyzed in logarithmic fold changes. The Behrens–Fisher approach was used for the 95% confidence intervals of group differences in levels of amino acids. The Appendix Supplementary Methods presents the construction of the permutation tests, pathway analysis, and vitamin A status markers.

## Data availability

All R code used in statistical analysis and building figures for this article are open access in GitHub: https://github.com/topihovinen/mirahelsinki.

This study includes no data deposited in external repositories.

**Expanded View** for this article is available online.

## Acknowledgements

We thank the families who participated in this study; Markus Innilä for the management of the samples; the City of Helsinki, especially the daycare personnel and Tarja Heikkinen from the City of Helsinki Service Center, for their collaboration; Dr. Jürgen Erhardt of VitMinLab for measuring and interpreting data related to vitamin A status, iron status, and inflammation. The authors also wish to thank the following funding sources: Sigrid Jusélius Foundation, Academy of Finland, Helsinki University Hospital and University of Helsinki (for AS), Finnish Cultural Foundation, and Finnish Society for Nutrition Research (for LK).

## Author contributions

AS, ME, RF, LK, PI, and TH devised the study concept and design. TH, LK, AS, ME, and RF drafted the manuscript. All authors critically revised the manuscript for important intellectual content. TH, LK, ES, and JN did the statistical analysis. LK, ES, RF, and TH measured the anthropometrics and conducted the urine sample collection. NZ acquired and preprocessed the untargeted metabolomics, HG conducted cholesterol biomarker measurements, and MN acquired targeted bile acid measurements. MN conducted the targeted bile acid, and HG the targeted cholesterol biomarker measurements. All authors analyzed and interpreted data, and provided administrative, technical, and material support. AS and TH had full access to original data, and all authors to the results. AS and ME supervised the study.

## Conflict of interest

TH, RF, ES, PI, MN, JN, and HG have nothing to disclose. LK was a board member of the company TwoDads at the time of the study. ME and LK disclose author's fee from Finnish Medical Journal Duodecim. NZ is the founder and scientific advisor of General Metabolics. AS has obtained speaker fees from Orion Pharma and is part of Khondrion SAB, unrelated to the current study.

## For more information

National Nutrition Council / Finnish Food Authority. Recommendations for vegan diet: https://www.ruokavirasto.fi/en/themes/healthy-diet/nutrition-and-food-recommendations/vegan-diet/

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
