## [Review Process File · EMBO Molecular Medicine]

Vegan diet in young children remodels metabolism and challenges the statuses of essential nutrients

Topi Hovinen, Liisa Korkalo, Riitta Freese, Essi Skaffari, Pirjo Isohanni, Mikko Niemi, Jaakko Nevalainen, Helena Gylling, Nicola Zamboni, Maijaliisa Erkkola, and Anu Suomalainen
DOI: [10.15252/emmm.202013492](https://doi.org/10.15252/emmm.202013492)

Corresponding author: Anu Suomalainen (anu.wartiovaara@helsinki.fi)

Review Timeline:

Submission Date:	24th Sep 20
Editorial Decision:	8th Oct 20
Revision Received:	18th Nov 20
Editorial Decision:	1st Dec 20
Revision Received:	3rd Dec 20
Accepted:	4th Dec 20

Editor: Zeljko Durdevic

Transaction Report:

8th Oct 2020

Dear Prof. Suomalainen,

Thank you for the submission of your manuscript to EMBO Molecular Medicine. We have now received feedback from the three reviewers who agreed to evaluate your manuscript. As you will see from the reports below, the referees acknowledge the interest of the study and are overall supporting publication of your work pending appropriate revisions. Please include vegetarian group in the statistical analysis and discuss its metabolic profile in relation to omnivore and vegan groups as suggested by the referees #2 and #3. Also, please provide rationale for selection of the alternate vitamin A status indicator instead of the standard serum retinol - re-evaluation of vitamin A status is not required.

Addressing the reviewers' concerns in full will be necessary for further considering the manuscript in our journal. EMBO Molecular Medicine encourages a single round of revision only and therefore, acceptance or rejection of the manuscript will depend on the completeness of your responses included in the next, final version of the manuscript. For this reason, and to save you from any frustrations in the end, I would strongly advise against returning an incomplete revision.

I look forward to reading a new revised version of your manuscript as soon as possible.

***** Reviewer's comments *****

Referee #1 (Remarks for Author):

This is an interesting paper very well conducted and it will be very useful to extend the study to include more children.

Only one error at line 62: table 2 instead of 1.

Please cite the following papers:

- 1) Baldassarre ME, Panza R, Farella I, Posa D, Capozza M, Mauro AD, Laforgia N. Vegetarian and Vegan Weaning of the Infant: How Common and How Evidence-Based? A Population-Based Survey and Narrative Review. *Int J Environ Res Public Health*. 2020 Jul 5;17(13):4835.
- 2) Farella I, Panza R, Baldassarre MI. The Difficult Alliance between Vegan Parents and Pediatrician: A Case Report. *Int. J. Environ. Res. Public Health* 2020, 17, 6380.

Referee #2 (Remarks for Author):

The study presents important work evaluating clinical nutrient status outcomes comparing young vegan children to vegetarian and omnivores recruited through daycare centers. Though the evidence shows some possible differences between groups, I do not think the paper provided sufficient evidence to support the overarching assertion that vegan children may have insufficient vitamin A and vitamin D status.

I do not think the method used to evaluate vitamin A status is sufficient to evaluate whether the vegan kids were insufficient or deficient in vitamin A. WHO and IOM recommend serum retinol concentration for defining vitamin A adequacy. The method used/cited was developed for resource-poor countries that do not have access to this established standard. I am wondering why the serum retinol concentrations were not evaluated or reported? I think methods like serum retinol and relative-dose-response test should be used to make conclusions about insufficiency in the absence of clinical signs of deficiency, especially when the sample size of the vegan group is so small. If it's possible to do a serum retinol test, that would add to the manuscript. Additionally, there is some evidence that in early life chylomicrons and other lipoproteins may be important for extrahepatic vitamin A distribution (see <https://www.jlr.org/content/55/8/1738.short>). It's also possible that distribution of vitamin A may not depend as much on RBP-transthyretin in young children.

There is no evidence of protein or zinc deficiency in the vegan kids in this study, so I don't think metabolism/synthesis of RBP is a relevant concern in this group of 6 vegan kids (line 184; and a relative-dose-response test may have shed some light on that issue). If there really is a vitamin A insufficiency problem in these 6 vegan kids, could a vitamin A insufficiency be a sign that beta-carotene alone is not sufficient to maintain adequate vitamin A status in vegan children? The authors report total RAE from both diet and supplements. Vegan children did not get vitamin A from supplements (page 32), but it is unclear whether vegan children included in this study had ANY retinoids in their diet. If the authors could parse out the RAE coming from retinoids vs. carotenoids in each of the diet groups and evaluate whether consumption of retinoids is associated with vitamin A status, I think that would add to the paper and discussion.

Clinically, 50 nmol/L 25(OH)D is sufficient, and the evidence presented here doesn't support that any participant in the study had insufficient vitamin D status. It's possible that vegan and vegetarian groups don't have optimal vitamin D status, but there is no analysis of PTH or discussion about optimization. Additionally the statistics presented in Table 3 (page 33) only show a comparison between the vegan vs. omnivore groups. The vitamin D status looks very similar between the vegan and vegetarian groups; why were statistical comparisons not reported across all three groups? I think if the vegetarian group is also different from the omnivore group, that information should be included and discussed.

Referee #3 (Remarks for Author):

In the manuscript "Vegan diet in young children remodels metabolism and challenges the statuses of essential nutrients", Hovinen and colleagues study the influences of vegan diet on nutritional status and metabolism in Finnish children. They compare the serum metabolomic profiles of vegan, vegetarian and omnivorous children and identify metabolites and metabolic pathways significantly different between vegan children and others. They find that vegan children have lower levels of vitamin A and D and alterations in the metabolism of cholesterol, fatty acids and amino acids, thereby raising concerns about detrimental health outcomes caused by nutritional deficiency related to adherence to vegan diet in early childhood. This is an important topic since the vegan

diet has been considered a 'healthy' diet for adults but how this diet affects health in children is less well understood.

Major:

- In the discussion authors conclude "The children who followed the vegan diet from birth showed a metabolic profile and nutrient status distinct from those of lacto-ovo-vegetarians and omnivores". However the statistical comparisons described in the text and shown in the figures were between omnivore and vegan, not vegetarian and vegan, and therefore it should not be concluded that the metabolic profile of vegans was distinct from vegetarians.

According to the figures 2, 3, and 4, it seems that vegetarian diet resulted in similar trends of changes in metabolism, although the effect sizes were smaller. Could the authors comment on the difference between vegetarian and vegan diets as revealed by their data? Is the vegetarian group in general closer to the group of omnivores or closer to the vegan group in terms of the nutritional status and metabolism? A PCA or clustering analysis might be helpful to illustrate this.

Minor:

- It would be useful to state when the blood samples to be used for metabolomics were collected (were the children fasting, was it after they ate their daycare meal, etc).
- It might be helpful to graph the main dietary intake differences between groups in Figure 1
- On line 102 it says that DHA was higher in vegans, but the Figure 4A shows that it is lower.
- Statistics should be included in Figure 4.

The Authors' Responses to the Referees' Comments

We would like to thank the Referees and the Editor for the overall positive reception and constructive comments on our manuscript. We have now considered all of the comments and resolved the concerns by the actions **marked by bold text** our detailed response below.

As a summary of the main changes made: 1) We have now included the vegetarian group in statistical analysis, in addition to vegan and omnivore groups: we conducted similar permutation tests combined with Benjamini-Hochberg corrections for multiple testing as a secondary statistical analysis to provide significances of vegetarian group compared to other diet groups. Additionally we ran a hierarchical clustering analysis to further explore the relations between diet groups and discussed the results accordingly. 2) We provided the rationale for choosing RBP as the vitamin A status indicator, as suggested by the Editor and referees #2 and #3.

In the manuscript text, changes based on the Referees' and Editor's comments have been marked with a **green background** to facilitate interpretation of our revisions. In addition, we rephrased some ambiguous expressions in the Discussion and added detail to Methods that had not been included in the first submission. These minor but reasonable improvements that are not directly related to comments below are highlighted in this final submission with **yellow background**. Please find our detailed responses below.

Referee #1:

Referee: This is an interesting paper very well conducted and it will be very useful to extend the study to include more children.

Authors: We would like to thank the Reviewer for favourable comments.

Ref: Only one error at line 62: table 2 instead of 1.

Au: This error **has now been corrected**.

Ref: Please cite the following papers:

1) Baldassarre ME, Panza R, Farella I, Posa D, Capozza M, Mauro AD, Laforgia N. Vegetarian and Vegan Weaning of the Infant: How Common and How Evidence-Based? A Population-Based Survey and Narrative Review. *Int J Environ Res Public Health*. 2020 Jul 5;17(13):4835.

2) Farella I, Panza R, Baldassarre MI. The Difficult Alliance between Vegan Parents and Pediatrician: A Case Report. *Int. J. Environ. Res. Public Health* 2020, 17, 6380.

Au: Thank you for suggesting these very recent publications related to our study. We have read the suggested literature and **cited the suggested publications in the introduction as a valuable addition to the literature**.

Referee #2:

Ref: The study presents important work evaluating clinical nutrient status outcomes comparing young vegan children to vegetarian and omnivores recruited through daycare centers.

Au: We would like to thank the Referee for appreciating the value of our study.

Ref: Though the evidence shows some possible differences between groups, I do not think the paper provided sufficient evidence to support the overarching assertion that vegan children may have insufficient vitamin A and vitamin D status.

Au: Thank you for the comment. To our opinion, the significantly lower and only borderline sufficient concentrations of total vitamin D (D2 + D3) and RBP in our vegan sample give sufficient evidence to raise concern on the increased risk of vitamin A or D insufficiency in general vegan child population, particularly during winter with limited exposure to sunlight. As insufficiencies for these vitamins are known to cause consequences for child health, we consider it important to make the point. This is the first pilot study on vegan children, and the results provide an important basis and point the way for further and targeted studies. **We have now revised the Discussion of our manuscript to ensure that our conclusions are cautious.** Please find more detailed arguments below.

Ref: I do not think the method used to evaluate vitamin A status is sufficient to evaluate whether the vegan kids were insufficient or deficient in vitamin A. WHO and IOM recommend serum retinol concentration for defining vitamin A adequacy. The method used/cited was developed for resource-poor countries that do not have access to this established standard. I am wondering why the serum retinol concentrations were not evaluated or reported? I think methods like serum retinol and relative-dose-response test should be used to make conclusions about insufficiency in the absence of clinical signs of deficiency, especially when the sample size of the vegan group is so small. If it's possible to do a serum retinol test, that would add to the manuscript. Additionally, there is some evidence that in early life chylomicrons and other lipoproteins may be important for extrahepatic vitamin A distribution (see <https://www.jlr.org/content/55/8/1738.short>). It's also possible that distribution of vitamin A may not depend as much on RBP-transthyretin in young children.

Au: We thank the Referee for the constructive criticism. We agree with the Referee that serum retinol would be the biomarker of choice for vitamin A analysis. However, the sample available from healthy infants and young children was small in volume, and we optimized sample use for the comprehensive analysis of metabolic and cofactor state, as such wide view to vegan infants has not been available before. To our best knowledge, RBP is considered a sufficient method to describe vitamin A status in non-obese population with little or no underlying medical diseases (<https://doi.org/10.3945/jn.115.229708>). In addition, we also included the alternative mathematical model of vitamin A status with RBP, CRP and transthyretin to account for confounding effects of potential unnoticed conditions of inflammation and protein deficiency. The significantly lower status of vitamin A was supported by both of these approaches. The literature cited by the Referee on vitamin A distribution in neonate rat pups indicates the complexity in interpretation of vitamin A status, but it is unclear how generalizable these mechanisms are for humans. However, we agree that serum retinol measurement and screening of clinical signs of vitamin A insufficiency should be the next important step in follow-up investigation of the vitamin A status of vegan children, now when a concern has been raised. **We now suggest this more clearly in the Discussion.**

Ref: There is no evidence of protein or zinc deficiency in the vegan kids in this study, so I don't think metabolism/synthesis of RBP is a relevant concern in this group of 6 vegan kids (line 184; and a relative-dose-response test may have shed some light on that issue).

Au: We agree with the Referee that our study shows no evidence of zinc deficiency in vegan children of our study, exactly as we reported in the manuscript. We found, however, low transthyretin levels and lowered protein intake, together with generally lowered essential amino acids, suggesting lower protein (or specific amino acid) status in vegans than in omnivores. On line 184, we discuss possible factors contributing to the observed low levels of RBP in vegan children. We fully share the view of the Referee that the roles of zinc

deficiency, iron deficiency and low energy intake on RBP levels are not supported by our data of biomarkers and intake. This further suggests that the findings on RBP reflect alterations in either vitamin A or protein status. **We have rephrased the sentences accordingly.**

Ref: If there really is a vitamin A insufficiency problem in these 6 vegan kids, could a vitamin A insufficiency be a sign that beta-carotene alone is not sufficient to maintain adequate vitamin A status in vegan children? The authors report total RAE from both diet and supplements. Vegan children did not get vitamin A from supplements (page 32), but it is unclear whether vegan children included in this study had ANY retinoids in their diet. If the authors could parse out the RAE coming from retinoids vs. carotenoids in each of the diet groups and evaluate whether consumption of retinoids is associated with vitamin A status, I think that would add to the paper and discussion.

Au: Thank you for the comment. The software and food composition database we used originally did not allow us to study retinoid and carotenoid intakes separately. However, we appreciate the importance of this point and have now 1) manually revised the consumed food items in our data of each participant in order to find which foods contained retinoids naturally or were fortified with retinoids, 2) checked the quantity of retinoids from product details online and food composition databases and 3) calculated the proportion of retinoids in the total RAE in the raw ingredients consumed by each diet group. We found that the only food items in food records of the vegan group containing any retinoids were margarines, contributing on average 33% of the total RAE in vegan group. The corresponding proportion in omnivores and vegetarians was 60% and 41%, respectively. **We have added this information in the Results and the detailed Appendix Supplementary Methods.**

Ref: Clinically, 50 nmol/L 25(OH)D is sufficient, and the evidence presented here doesn't support that any participant in the study had insufficient vitamin D status. It's possible that vegan and vegetarian groups don't have optimal vitamin D status, but there is no analysis of PTH or discussion about optimization.

Au: Thank you for the valid comment. We agree that the claim on vitamin D showing “compromised levels” is not supported by our data. **We have rephrased the sentence to “markedly low”** which reflects better the significant difference to omnivores and lowest median of all groups without indicating insufficiency. PTH was not analyzed, but certainly a good idea to analyse it in follow-up studies. **We have also rephrased the ambiguous use of deficiency in the discussion concerning vitamin D levels in Finland** – insufficiency, defined by low measured levels of vitamin D, is commonplace in Finnish population at wintertime while deficiency, manifesting as rickets or other major complications, is not.

Ref: Additionally the statistics presented in Table 3 (page 33) only show a comparison between the vegan vs. omnivore groups. The vitamin D status looks very similar between the vegan and vegetarian groups; why were statistical comparisons not reported across all three groups? I think if the vegetarian group is also different from the omnivore group, that information should be included and discussed.

Au: We agree with the point. To follow our original research plan on comparing vegans to omnivores, we have been very strict in classifications, as even single portions of animal-derived foods excluded the subjects from vegan group, and therefore the vegetarian group is quite heterogeneous from almost vegan to those who regularly consume dairy, eggs and fish. In addition, our vegetarian group may not represent general vegetarian population very well – the vegetarian children in our study consumed a notable amount of fully vegan meals. Keeping this in mind, rather interestingly we still see a roughly gradual change in many of the measured biomarkers when proceeding from omnivores to vegetarians to vegans that we wanted to include in the figures. **We have now included results from permutation tests**

comparing vegetarian group to other diet groups, and a hierarchical clustering analysis (as suggested by Ref #3) as a secondary statistical analysis in all Tables and Figures, and discussed the vegetarian group in more detail, addressing also the abovementioned issues in Discussion.

Referee #3:

Ref: In the manuscript "Vegan diet in young children remodels metabolism and challenges the statuses of essential nutrients", Hovinen and colleagues study the influences of vegan diet on nutritional status and metabolism in Finnish children. They compare the serum metabolomics profiles of vegan, vegetarian and omnivorous children and identify metabolites and metabolic pathways significantly different between vegan children and others. They find that vegan children have lower levels of vitamin A and D and alterations in the metabolism of cholesterol, fatty acids and amino acids, thereby raising concerns about detrimental health outcomes caused by nutritional deficiency related to adherence to vegan diet in early childhood. This is an important topic since the vegan diet has been considered a 'healthy' diet for adults but how this diet affects health in children is less well understood.

Au: We thank the referee for the positive reception of our manuscript.

Ref: Major:

- In the discussion authors conclude "The children who followed the vegan diet from birth showed a metabolic profile and nutrient status distinct from those of lacto-ovo-vegetarians and omnivores". However the statistical comparisons described in the text and shown in the figures were between omnivore and vegan, not vegetarian and vegan, and therefore it should not be concluded that the metabolic profile of vegans was distinct from vegetarians.

According to the figures 2, 3, and 4, it seems that vegetarian diet resulted in similar trends of changes in metabolism, although the effect sizes were smaller. Could the authors comment on the difference between vegetarian and vegan diets as revealed by their data? Is the vegetarian group in general closer to the group of omnivores or closer to the vegan group in terms of the nutritional status and metabolism? A PCA or clustering analysis might be helpful to illustrate this.

Au: To follow our original research plan on comparing vegans to omnivores, we have been very strict in classifications, as even single portions of animal-derived foods excluded the subjects from vegan group, and therefore the vegetarian group is quite heterogeneous from almost vegan to those who regularly consume dairy, eggs and fish. The vegetarian children in our study consumed a notable amount of fully vegan meals, as they enjoyed vegan diet in daycare. Keeping this in mind, rather interestingly we still see a roughly gradual change in many of the measured biomarkers when proceeding from omnivores to vegetarians to vegans that we wanted to include in the figures. **We have now included the statistics to the Tables and Figures, and noted in the Discussion that our vegetarian group is heterogeneous but enjoyed vegan meals in daycare.**

Thank you for the excellent suggestion to apply clustering analysis. We conducted clustering analysis with Ward method and Euclidean distance, and the results **have now been included in our manuscript Methods, Results, Discussion and as Figure EV1**. To further study the relations between all three groups, we included the results from permutation tests comparing vegetarians with other diet groups as a secondary statistical analysis (see also the

last comment of Referee #2). Both analyses supported our claim on vegan group being distinct from the vegetarian group of our study.

Ref: Minor:

- It would be useful to state when the blood samples to be used for metabolomics were collected (were the children fasting, was it after they ate their daycare meal, etc).

Au: Thank you for the valid suggestion. We paid careful attention to this in the study design. The blood samples were collected after overnight fasting, as advised for the parents. We have now **included information of the participant state at time of blood and urine sample collection in Methods.**

Ref: - It might be helpful to graph the main dietary intake differences between groups in Figure 1

Au: Thank you for the excellent suggestion. **We have now included the main dietary intake differences in Figure 1 below the anthropometric data.**

Ref: - On line 102 it says that DHA was higher in vegans, but the Figure 4A shows that it is lower.

Au: Thank you for noticing this mistake in text, **which has now been corrected.**

Ref: - Statistics should be included in Figure 4.

Au: **We have now included the p-values from applicable statistical analyses in Figure 4.**

References:

Tanumihardjo SA, Russell RM, Stephensen CB, Gannon BM, Craft NE, Haskell MJ, Lietz G, Schulze K, Raiten DJ (2016) Biomarkers of Nutrition for Development (BOND) – Vitamin A Review. *J Nutr* 146(9): 1816S-1848S. <https://doi.org/10.3945/jn.115.229708>

1st Dec 2020

Dear Prof. Suomalainen,

Thank you for the submission of your revised manuscript to EMBO Molecular Medicine. I am pleased to inform you that we will be able to accept your manuscript pending the following final amendments:

- 1) Tables: Please include Table 1 to the manuscript file and make Table 2 and 3 to EV tables and add their legends to the manuscript text file.
- 2) In the main manuscript file, please do the following:
 - Correct/answer the track changes suggested by our data editors by working from the attached/uploaded document.

The authors performed the requested changes.

The authors performed the requested changes.

Corresponding Author Name: Anu Suomalainen

Manuscript Number: EMM-2020-13492